# Preventive Role of Diet Interventions and Dietary Factors in Type 2 Diabetes Mellitus: An Umbrella Review

**DOI:** 10.3390/nu12092722

**Published:** 2020-09-06

**Authors:** Phung Lam Toi, Thunyarat Anothaisintawee, Usa Chaikledkaew, Jamaica Roanne Briones, Sirimon Reutrakul, Ammarin Thakkinstian

**Affiliations:** 1Mahidol University Health Technology Assessment (MUHTA) Graduate Program, Mahidol University, Bangkok 10400, Thailand; phunglamtoi@hspi.org.vn (P.L.T.); usa.chi@mahidol.ac.th (U.C.); jambriones@gmail.com (J.R.B.); ammarin.tha@mahidol.edu (A.T.); 2Health Strategy and Policy Institute, Ministry of Health, Hanoi 10400, Vietnam; 3Department of Family Medicine, Ramathibodi Hospital, Mahidol University, Bangkok 10400, Thailand; 4Social Administrative Pharmacy Division, Department of Pharmacy, Faculty of Pharmacy, Mahidol University, Bangkok 10400, Thailand; 5Division of Endocrinology, Diabetes and Metabolism, Department of Medicine, University of Illinois College of Medicine at Chicago, 835 S Wolcott, Ste E625, Chicago, IL 60612, USA; sreutrak10800@gmail.com; 6Department of Clinical Epidemiology and Biostatistics, Ramathibodi Hospital, Mahidol University, Bangkok 10400, Thailand

**Keywords:** diabetes mellitus, dietary patterns, nutrient, food, systematic review

## Abstract

Background: Although the body of evidence indicates clear benefits of dietary modifications for prevention of type-2 diabetes mellitus (T2DM), it may be difficult for healthcare providers to recommend which diet interventions or dietary factors are appropriate for patients as there are too many modalities available. Accordingly, we performed an umbrella review to synthesize evidence on diet interventions and dietary factors in prevention of T2DM. Methods: Medline and Scopus databases were searched for relevant studies. Systematic reviews with meta-analyses of randomized-controlled trial or observational studies were eligible if they measured effects of diet interventions and/or dietary factors including dietary patterns, food groups, and nutrients on risk of T2DM. The effect of each diet intervention/factor was summarized qualitatively. Results: Sixty systematic reviews and meta-analyses were eligible. Results of the review suggest that healthy dietary patterns such as Mediterranean and Dietary Approaches to Stop Hypertension (DASH) diets, and high consumption of whole grains, low-fat dairy products, yogurt, olive oil, chocolate, fiber, magnesium, and flavonoid significantly reduced the risk of T2DM. In contrast, high glycemic index and glycemic load diets, high consumption of red and processed meat, and sugar or artificial sugar-sweetened beverages significantly increased risk of T2DM. Prescribing diet interventions with or without physical activity interventions significantly decreased risk of T2DM in both high-risk and general population. Conclusion: High consumption of Mediterranean and DASH diet, and interventions that modified the quality of diet intake significantly reduced risk of T2DM especially in the high-risk population. These lifestyle modifications should be promoted in both individual and population levels to prevent and decrease burden from T2DM in the future.

## 1. Introduction

Type-2 diabetes mellitus (T2DM) is a major public health problem. Around 425 million people globally were diagnosed with T2DM in 2017 and this is predicted to increase to 629 million by 2045 [1]. T2DM is also a significant risk factor of cardiovascular diseases, the leading cause of morbidity and mortality in population worldwide [2]. Moreover, the economic burden on T2DM contributes to approximately 12% of global health expenditure from diabetic treatment and its complications [1]. As such, there is an urgency to decrease the global health and economic burden related to T2DM.

Genetic and lifestyle factors, especially diet, are significant determinants of T2DM. Lifestyle factors are modifiable and thus should be the focus in the effort to lower T2DM risk [3]. Several systematic reviews and meta-analyses (SRMA) of randomized controlled trials (RCT) [4,5,6] indicated that interventions that modified diet quality (e.g., low calories and low-fat diet) and/or enhanced level of physical activity can delay or prevent the onset of T2DM. However, the evidence focused only on high-risk population such as people with prediabetes or obesity. In the general population, evidence on the benefit of lifestyle factors were mostly based on observational studies that measured the association between lifestyle factors (e.g., dietary factors and level of physical activity) and risk of T2DM. Dietary factors can be divided into many sub-categories such as dietary patterns (e.g., Mediterranean and Dietary Approaches to Stop Hypertension (DASH) diet), food groups (e.g., nuts, whole grains, vegetables, fruits), and food nutrients. Several SRMAs investigated the effect of dietary factors on risk of T2DM. However, some of this evidence shows conflicting results. For instance, three SRMAs [7,8,9] assessing the effect of a high intake of dairy products demonstrated significant benefits for T2DM prevention, whereas the other 2 SRMAs [10,11] showed non-significant benefits.

Therefore, a comprehensive review is essential to summarize the effect of dietary factors and diet interventions on T2DM risk. In addition, a thorough review of the quality of previous evidence is necessary before making a strong conclusion about the benefit of dietary factors and diet interventions in the prevention of T2DM. As such, we performed an umbrella review aiming to comprehensively summarize all available evidence of the effect of dietary factors (e.g., dietary patterns, food groups, and nutrients) and diet interventions on the risk of T2DM in both general and high-risk population. The results from this review should be beneficial for health care providers and policymakers in applying them in clinical and public health practices.

## 2. Materials and Methods

The methodology of this umbrella review was done in adherence to the proposal registered in PROSPERO (CRD42018105292). This review was also conducted according to the preferred reporting items for systematic reviews and meta-analyses (PRISMA) guidelines [12].

### 2.1. Location of Studies

PubMed and Scopus databases were searched for relevant studies from inceptions until August 2020. Search terms and search strategies for each database are described in Appendix A. Reference lists of included systematic reviews were explored. We also contacted experts in this field for potentially relevant studies.

### 2.2. Selection of Studies

Two authors (P.L.T. and J.R.B.) independently selected studies. Percent agreement between two reviewers was 92.68% with kappa statistic of 0.83. SRMAs of randomized controlled trials (RCT) or observational studies published in any language were eligible if the study’s participants were nondiabetic adults aged greater than 18 years, exposures or interventions were dietary factors (e.g., dietary patterns, food groups, and nutrients), diet intervention, or combined lifestyle interventions that included diet intervention, and had evaluated the incidence of T2DM for the outcome. Studies were excluded if the study compared diet interventions with pharmacological interventions.

### 2.3. Data Extraction

Two authors (P.L.T. and T.A.) independently extracted data including general information, review methodology (i.e., review question according to participants (P), interested intervention/exposure (I), comparator (C), interested outcome (O), and study design of included studies, number of databases used for searching, last searched date, number of included studies), characteristics of included studies (setting, participants, sample size, interventions/comparators, and outcomes), and findings (i.e., pooled risk ratio (RR), odds ratio (OR), and hazard ratio (HR), along with their 95% confidence intervals (CI), heterogeneity, and publication bias). Disagreement of data extraction between the two reviewers was resolved by discussion with a third-party reviewer (U.C., A.T.).

### 2.4. Risk of Bias Assessment

The quality of SRMAs was assessed by Assessing the Methodological Quality of Systematic Reviews-2 (AMSTAR-2) [13], instead of A Risk of Bias Assessment Tool for Systematic Reviews (ROBIS) as stated in the review proposal, because the AMSTAR-2 is a more updated tool that applies to SRMA of non-RCTs [13].

AMSTAR-2 involves critical and non-critical domains. The critical domains consist of seven items (i.e., protocol registration, adequacy of literature search, justification of excluding studies, performing risk of bias assessment, appropriateness of meta-analytical method, consideration of risk of bias when interpreting the results, and assessment of publication bias). The rest of the nine items are non-critical domains. Each item was graded as yes, partial yes, and no, if the SRMA completely, partially, and did not comply with that criteria, respectively. The overall quality of the SRMA was then graded as “high confidence” if the SRMA complied with all items of critical and noncritical domains, or only one noncritical domain was violated; “moderate confidence” if the SRMA complied with all critical domains but more than one noncritical domains were violated; “low and critically low confidence,” if one and more than one critical domains were violated regardless noncritical domain [13]. Two authors (P.L.T., T.A.) independently assessed the quality of each study and discrepancies between the two authors were solved by consensus and discussion (U.C., A.T.).

### 2.5. Data Synthesis

The characteristics of the included SRMAs were qualitatively summarized. In addition, specific characteristics of meta-analysis were also described including type of meta-analysis (direct or network meta-analyses), parameter used for pooling, results of heterogeneity between studies and publication bias assessment.

The study matrix was constructed considering individual included studies of each SRMA in row across each meta-analysis in columns. Each study may be included in each meta-analysis more than once, thus form overlapping evidence. A degree of overlapping was therefore assessed by calculation of the corrected covered area (CCA) for each type of intervention [14]. The CCA ranging from 0% to 5% suggested the slight degree of overlapping, while CCA ranging from 6-10%, 11-15%, and greater than 15% indicated the moderate, high, and very high overlapping, respectively. Furthermore, an excess significant finding (ESF) was estimated [15] to assess if the individual meta-analyses were over-claimed. Finally, forest plots of the effects of dietary factors and diet interventions from good quality SRMAs or the most recent ones were constructed.

STATA 15.0 (Stata Corp, College Station, TX, USA) was utilized for constructing forest plot as well as in measuring excess significant finding.

## 3. Results

A total of 3408 articles were identified which yielded 60 eligible SRMAs for the umbrella review (Figure 1). Characteristics of included SRMAs are summarized in Appendix A. These studies were published between 2005 and 2020. The number of databases used ranged from 1 to 13 with a median of 3. Number of included individual studies ranged from 3 to 88 with a median of 10 studies.

SRMAs were mainly conducted in Europe (21/60), followed by Asia (16/60), U.K (10/60), and the U.S. (10/60). Government was the main source of funding (27/60), whereas 18% (11/60) of included SRMAs did not report the funding source. Most of SRMAs reported no conflict of interest (48/60). All SRMAs applied pairwise meta-analysis for pooling effect sizes, except one that utilized a network meta-analysis of RCT [16]. The diet interventions and dietary factors were re-categorized into five groups, namely dietary patterns, food groups, food nutrients, diet interventions, and combined diet and physical activity interventions. Eighteen studies were SRMAs of RCTs, while forty-two studies were SRMAs of observational studies. Studies that considered diet interventions and combined diet and physical activity interventions were only SRMAs of RCTs, while almost all studies considering dietary factors as interested exposures were SRMAs of cohort studies.

### 3.1. Dietary Factors

Forty-three SRMAs assessed the effects of dietary factors. Dietary factors were classified into food groups, food nutrients, and dietary patterns. Characteristics of the included SRMAs are presented in Appendix A.

#### 3.1.1. Food Groups

Twenty-four SRMAs of cohort studies assessed the effect of food groups on risk of T2DM. Study’s participants were the general population for all SRMAs. The effects of each food group are presented in Figure 2 and Appendix A.

##### Fruits and Vegetables

Results from four SRMAs [17,18,19,20] found that compared to low intake, high fruit and vegetable intake did not significantly lower the risk of T2DM with pooled RRs ranging from 0.93 (95% CI: 0.87,1.00) to 1 (95% CI: 0.92,1.09), see Figure 2A. When considering vegetable and fruit separately, high vegetable intake did not significantly lower the risk of T2DM, but the evidence for high fruit intake were inconsistent. Two SRMAs found non-significant benefits of high fruit intake, while one SRMA suggested that high fruit intake significantly decreased the risk of T2DM with pooled RR of 0.92 (95% CI: 0.86, 0.97) [21].

##### Whole and Refined Grains, Nuts, and Legumes

Results from four SRMAs of cohort studies [9,22,23,24] found that high whole-grain intake significantly decreased the risk of T2DM with pooled RR ranging from 0.74 (95% CI: 0.69, 0.80) to 0.79 (95% CI: 0.72, 0.87), see Figure 2A. In addition, every 30 g/day increase in an intake of whole-grain significantly reduced the risk of T2DM around 13% (pooled RR = 0.87; 95% CI: 0.82, 0.93) [9], see Figure 2B. In contrast to whole grains, results from two SRMAs [9,23] suggested that a high intake of refined grains was not associated with risk of T2DM.

For nuts and legume intake, both highest versus lowest and dose-response analyses (e.g., every 4 servings per week or 50 g per day increase in consumption of legumes) did not suggest a significant benefit of legumes for T2DM prevention [9,25,26] (Figure 2B), while evidence for nuts intake was inconsistent. Findings from both highest versus lowest and dose-response analysis of Schwingshackl et al. [9] suggested no benefits of nuts for prevention of T2DM, whereas results from Afshin et al. indicated that every four servings per week increase in intake of nuts significantly lowered risk of T2DM with pooled RR of 0.87 (95% CI: 0.81, 0.94) [26].

##### Dairy Products

Three SRMAs of cohort studies [8,10,11] assessed the effect of high-fat and low-fat dairy products, milk, cheese, and yogurt on the risk of T2DM. For high and low-fat dairy products, every 200 g per day increase in an intake or high intake of low-fat dairy products significantly decreased risk of T2DM with pooled RRs ranging from 0.83 (95% CI: 76, 0.90) to 0.96 (95% CI: 0.92, 1.00), while high intake of high-fat dairy products and milk were not associated with the risk of T2DM. A high intake of cheese and yogurt significantly lowered the risk of T2DM (Figure 2A).

##### Fish and Meat

Results from three SRMAs [9,27,28] did not reflect the benefit of high fish intake in lowering the risk of T2DM (Figure 2A). Finding from dose response analysis also indicate the non-significant benefit of regular consumption of fish in prevention of T2DM (Figure 2B).

Two SRMAs of cohort studies [9,29] assessed the association between red and processed meat consumption and the risk of T2DM. These two SRMAs found that high intake of red and processed meat significantly increased the risk of T2DM with pooled RR of 1.21 (95% CI: 1.13, 1.30) for red meat and pooled RR ranging from 1.27 (95% CI: 1.20, 1.35) to 1.41 (95% CI: 1.25, 1.60) for processed meat. In addition, every 100 g/day increase in intake of red meat and 50 g/day increase in intake of processed meat also significantly increased the risk of T2DM with pooled RRs of 1.17 (95% CI: 1.08, 1.26) and 1.37 (95% CI: 1.22, 1.55), respectively.

##### Egg, Olive Oil, Chocolate

Results from one SRMA of cohort [9] studies suggest that high intake of egg did not significantly decrease the risk of T2DM but high and every intake of olive oil 10 g/day significantly lowered the risk of T2DM with pooled RRs of 0.84 (95% CI: 0.77, 0.92) and 0.91 (95% CI: 0.87, 0.95), respectively (Figure 2A,B). One SRMA of cohort studies [30] also found that high and every two servings/week increase in intake of chocolate had significant benefits for the prevention of T2DM (Figure 2 and Appendix A).

##### Sugar Sweetened Beverages and Coffee

Two SRMAs of cohorts [9,31] assessed the effects of sugar and artificial sugar-sweetened beverages on risk of T2DM and one SRMA [32] investigated the effect of coffee drinking.

Both dose response and high versus low analyses suggested that intake of sugar sweetened beverages was associated with increased risk of T2DM with pooled RRs of 1.31 (95% CI: 1.20, 1.40) for high intake and 1.21 (95% CI: 1.12–1.31) for every intake of 250 mL/day [9]. Furthermore, every 1 serving/day increase in the consumption of artificial SSB significantly increased the risk of T2DM with the pooled RR of 1.25 (1.18–1.33) [31]. By contrast, coffee drinking was associated with 35% reduction in risk of T2DM when compared highest versus lowest consumption with pooled RR of 0.65 (95% CI: 0.54–0.78) [32] (Figure 2A,B).

#### 3.1.2. Food Nutrients

Eleven SRMAs of cohort studies and one SRMA of RCTs evaluated the effect of food nutrients on risk of T2DM. Almost all SRMAs were conducted in general population, and one SRMA of RCTs was conducted in patients with hypertension or cardiovascular diseases. Food nutrients were divided into fiber (3 SRMAs), magnesium (4 SRMAs), flavonoid (2 SRMA), and n-3 polyunsaturated fatty acid (PUFA) (3 SRMAs). Characteristics of eight SRMAs are presented in Appendix A and the effect of each food nutrients on the risk of T2DM are presented in Appendix A and Figure 3.

##### Fiber

Findings from two SRMAs indicated that a high intake of fiber significantly reduced the risk of T2DM with pooled RRs ranging from 0.81 (95% CI: 0.73, 0.90) [33] to 0.85 (95% CI: 0.77, 0.94) [34], see Figure 3A. Results from dose-response analysis also found that every 10 g/day increase in intake of fiber significantly reduced the risk of T2DM around 9% (95% CI: 4%, 13%) [33], see Figure 3B. However, when assessed the effect of high fruit and vegetable fiber intake, there was no significant benefit of both fruit and vegetable fiber in the prevention of T2DM [33,34,35] but a high intake of cereal fiber significantly decreased the risk of T2DM (Appendix A).

##### Magnesium, Flavonoid, n-3 PUFA

Four [35,36,37,38] and one [39] SRMA found that high intake of magnesium and flavonoid significantly decreased the risk of T2DM with pooled RRs ranging from 0.77 (95% CI: 0.72, 0.84) to 0.83 (95% CI: 0.80, 0.86) for high intake and pooled RR of 0.89 (95% CI: 0.82, 0.96) for dose response analysis (Figure 3A,B). However, one SRMA that assessed the effect of flavan-3-osl, a subclass of dietary flavonoid did not find the significant benefit of this subclass for the prevention of T2DM [40]. According to n-3PUFA, results from all three SRMAs did not show the significant benefit of a high intake of n-3 PUFA in prevention of T2DM [41,42,43].

#### 3.1.3. Dietary Patterns

Dietary patterns were eating styles, including Mediterranean diet, DASH diet, low glycemic index diet, or diet with high healthy eating index (HEI), or alternative healthy eating index (AHEI). The effects of dietary patterns were assessed by 10 SRMAs of cohorts (see Figure 4, Appendix A). Among them, three SRMAs [44,45,46] indicated that high glycemic index and glycemic load diets were associated with a higher risk of developing T2DM, with the pooled RRs of 1.08 (95% CI: 1.02, 1.15) to 1.16 (95% CI: 1.06, 1.26). This result was similar in either highest versus lowest or dose-response analyses. In contrast, Mediterranean (*n* = 4) and DASH diet showed protective effects on T2DM with the pooled RRs ranging from 0.77 (95% CI: 0.66, 0.89) [47] to 0.87 (95% CI: 0.82, 0.93) [48] and 0.73 (95% CI: 0.65, 0.83) [49] to 0.79 (95% CI: 0.66, 0.95) [50], respectively. Likewise, diet with high HEI and AHEI [48,50] also showed protective effects with the pooled RRs of 0.82 (95% CI: 076, 0.88) to 0.82 (95% CI: 0.74, 0.92) [48] and 0.77 (95% CI: 0.68, 0.86) to 0.79 (95% CI: 0.70, 0.89), respectively. Four SRMAs [48,49,50,51] quantified the effects of adherence to a healthy diet which was classified by factor analysis or principal component analysis. Pooled RRs varied from 0.86 (95% CI: 0.82, 0.90) to 0.78 (95% CI: 0.72, 0.85).

#### 3.1.4. Overlapping of Studies and Excess Significant Finding in Dietary Factors

The number of included studies in SRMAs for each intervention are described in Appendix A. A degree of study overlapping or corrected covered area (CCA) of each SRMA was estimated, which ranged from 6.2% to 11.7%. CCAs were moderate for pooling of food group (CCA = 6.2%) and dietary patterns (CCA = 9.7%); and high for food nutrients (CCA = 11.7%).

The excess significant finding (ESF) was calculated in 11 comparisons of exposure. Seven out of 11 comparisons showed evidence of ESF (Appendix A). In which, 4 and 3 comparisons were in dietary patterns and food groups, respectively.

### 3.2. Diet Intervention

Four SRMAs of RCTs assessed the effect of diet interventions [16,52,53,54] on T2DM risk. Characteristics of these SRMAs are described in Appendix A. Only people at risk such as those with impaired glucose tolerance (IGT), impaired fasting glucose (IFG), or obesity were included in the studies of diet interventions [16,53]. Details of diet intervention prescribed in the RCTs are described in Appendix A. Diet interventions in most studies were low calories diet (i.e., 25–30 kcal/kg or carbohydrate <120 g/day), and American Heart Association’s Step II diet (defined as fat intake of ≤30% of total energy, dietary saturated fat to <10% of energy and cholesterol to <300 mg/day). None of the additional behavioral intervention or supporting device was applied to these diet interventions. The duration of follow-up ranged from 1.5 to 9.4 years. Results of the included SRMAs suggest that dietary interventions could significantly lower the risk of T2DM in a high-risk population with pooled RRs ranging from 0.51 (95% CI: 0.39, 0.68) to 0.71 (95% CI: 0.55, 0.90) (see Figure 4A, Appendix A).

### 3.3. Combination of Diet and Physical Activity Interventions

Seventeen SRMAs of RCTs [3,4,5,6,16,52,53,54,55,56,57,58,59,60,61,62,63] reported the effects of combined diet and physical activity interventions in the prevention of T2DM. Diet interventions were mainly low calories and carbohydrates, high fiber, or diet control in general. Physical activity, mainly supervised by healthcare professionals, included aerobic exercise, resistance training, or encouragement to increase physical activity. Additionally, pedometers were also used to monitor compliance [3,58,59,64] (Appendix A). The duration of intervention and follow-up ranged from 6 months to 23 years. Most meta-analyses focused on high-risk people including obesity, metabolic syndrome, IGT, IFG, and women with a history of gestational diabetes mellitus (GDM). Included studies were conducted in various settings, with two reviews focusing in Asian population [58,62] (Appendix A).

Combined effect of diet and physical activity interventions significantly decreased risk of T2DM in both general and high-risk population with pooled RRs ranging from 0.43 (95% CI: 0.35, 0.52) to 0.55 (95% CI: 0.44, 0.70) for general population [52,58]. For obese people, pooled RRs was 0.44 (95% CI: 0.36, 0.52) [6] and for people with high-risk of CVDs pooled RRs ranging from 0.35 (95% CI: 0.14–0.85) [3] to 0.59 (95% CI: 0.52, 0.66) [56]. For people with prediabetes (e.g., impaired fasting glucose and impaired glucose tolerance test), combined diet and physical activity interventions significantly decreased the risk of T2DM with pooled RRs ranging from 0.39 (95% CI: 0.28, 0.53) to 0.65 (95% CI: 0.56, 0.75). (Figure 4B, Appendix A). Results were inconsistent in women with a history of GDM. A study by Goveia et al. [60] did not find significant benefits of combined diet and physical activity interventions, but findings from Li et al. [61] indicated that combined interventions significantly decreased the risk of T2DM around 43% (pooled RR = 0.57; 95% CI: 0.42, 0.78).

The overlapping among the included meta-analyses of combined diet and physical activity interventions was moderate (10.6%) (Appendix A). The ESF was calculated in 14 comparisons, in which 13 comparisons showed evidence of excess significant finding (Appendix A).

### 3.4. Methodological Quality of Studies

Results of quality assessment are presented in Appendix A. Confidence in the findings of included SRMAs are critically low, low, moderate, and high at 81.7%, 15.0%, 1.7%, and 1.7%, respectively. Most of SRMAs that had critically low confidence of their findings because they did not adhere to these following critical domains—(1) did not mention established review methods before conducting the review (80%), (2) did not provide the list and justification of excluded studies (86%), and (3) did not account for the risk of bias in individual studies when interpreting results (94%). Moreover, most reviews did not report funding sources of included studies as well as justified the study design included in the reviews.

## 4. Discussion

This umbrella review provides a summary of dietary factors, diet intervention, and combined diet and physical activity interventions and their effects on the risk of T2DM. We found that: (1) diet and combined diet and physical activity interventions significantly decreased the risk of T2DM in both general and high-risk population; (2) dietary patterns such as Mediterranean and DASH diets, a diet with high HEI, and AHEI scores were also beneficial in the prevention of T2DM; (3) among various dietary factors, only a couple of food groups and specific nutrients demonstrated the beneficial effects such as whole grain, low-fat dairy product, cheese, yogurt, olive oil, total fiber, dietary magnesium, and flavonoids; (4) unhealthy diets such as high consumption of high glycemic index and glycemic load diet, red meat and processed meat, sugar and artificial sweetened beverages can accelerate the development of T2DM.

Diabetes develops progressively as a result of the complex interaction between insulin resistance and beta cells dysfunction. Dietary inputs influence the glucose-insulin homeostasis and thus also affect the level of blood sugar. For instance, low-calories and low carbohydrate diets that aim to reduce body weight, a well-known associated factor of insulin resistance had beneficial effects on insulin sensitivity and blood sugar level [65,66]. Our review supported this hypothesis as our findings found that dietary interventions that were low calories and low carbohydrate diets (i.e., 25–30 kcal/kg or carbohydrate <120 g/day) significantly lower the risk of T2DM in both general and high-risk population. In the pathogenesis of T2DM, both the amount of carbohydrates and also the quality of carbohydrates play an important role. Good quality carbohydrates such as diet with low glycemic index or high complex carbohydrates are associated with increased insulin sensitivity [67] and improved beta-cell function [68]. These have been supported by the results from our review where diets with low glycemic index significantly decreased the risk of T2DM. In addition to macronutrients, micronutrients such as fiber also affect the glucose-insulin homeostasis in human bodies. High fiber intake can improve whole-body insulin sensitivity by increasing colonic production of the short-chain fatty acid acetate, propionate, and butyrate that are the significant end products of dietary fiber fermentation by the gut bacteria [69,70]. Several clinical trials found that high fiber intake prevents weight gain [71] and improves markers of insulin sensitivity [72]. Findings from our review correspond with these evidence that increasing fiber intake significantly decreased the risk of T2DM.

Although some food nutrients such as dietary fiber, flavonoids, and magnesium had a significant benefit for T2DM prevention, recent dietary recommendations focus on overall dietary patterns rather than single isolated nutrients because of the limitations of single-nutrient component approach. For instance, matrix of foods, food processing, and food preparation is very complicated and can strongly modify the food nutrients that finally impact the health effects. In addition, translation of nutrient-based recommendations to the public is difficult because the accurate estimation of food nutrients is too complex for the general population.

Dietary patterns, or combination of regularly consumed foods, can produce synergistic health effects not only by decreasing calories but also through metabolic propensity toward abdominal adiposity particularly visceral fat, which in turn influences the insulin response [73]. In addition, evidence also showed that dietary interventions such as low carbohydrate and low glycemic index, low fat, and low consumption of sugar sweetened beverages can reduce body weight and visceral fat [74,75]. Furthermore, a diet containing low carbohydrate and low glycemic index illustrated the improvement in beta cells function in IGT patients [76].

American Diabetes Association guideline recommends a dietary strategy for subjects with prediabetes such as Mediterranean diet, low-calories, and low-fat eating diet [77]. These eating styles can be translated into specific food groups. Mediterranean diet, for example, involving high in vegetables, fruits, legumes, nuts, beans, cereals, grains, fish, and unsaturated fats such as olive oil. Also, it usually includes a low intake of meat and dairy foods [78]. ADA guideline also recommends that the overall quality of food consumed (as measured by the AHEI), with an emphasis on whole grains, legumes, nuts, fruits and vegetables, and minimal refined and processed foods, is also important [77]. In this study, we confirm this by showing the protective effects of healthy eating styles (e.g., Mediterranean diet, DASH diet, and diet with high HEI, and AHEI index) and multiple food groups. We have pointed out that some particular types of “healthy foods” showing benefits such as green leafy vegetables and cruciferous vegetables. This result does not necessarily mean that only these food groups are beneficial. However, it does suggest that there is substantial uncertainty about other food groups and further research is needed to identify new food groups which are beneficial for prevention of disease.

Physical activity can slow or delay the progression of T2DM by directly increase insulin sensitivity and indirectly strengthen weight control. Moderate to vigorous-intensity physical activity improves beta cells function and glucose regulation, independent of obesity [79,80,81]. ADA recommends people at risk of T2DM should increase the physical activity to at least 150 min/week of moderate-intensified activity (e.g., brisk walking) [77]. Our findings show that combined diet and physical activity interventions significantly lowered risk of T2DM and the effect of combined interventions might be superior than the effect of diet intervention alone.

Sugar sweetened beverages significantly increased the risk of T2DM. Recently, artificial sugar sweeteners have emerged as an alternative for sugar sweetened beverages. However, findings from human trial suggested that continuous exposure to artificial sugar sweeteners could reduce acute insulin response, decrease insulin sensitivity, and enhance GLP-1 release in healthy subjects [82]. In addition, recent evidence showed that artificial sugar sweeteners have small beneficial effects on decreasing body mass index and fasting blood glucose [83]. Our review also found that artificial sugar sweetened beverages were associated with higher risk of T2DM. However, this finding comes from only one meta-analysis of cohort study. Future research, therefore, needs to strengthen this claim. We also found that higher consumption of coffee was shown to reduce 35% of the risk of T2DM compared to lower consumption. Again, this should be reconfirmed by further researches as there are variations in coffee types as well as the way of making coffee in which whether sugar or milk was added.

### Strengths and Limitations

Our study was the first umbrella review systematically summarizing broad evidence of dietary factors, diet intervention, and combined diet and physical activity interventions in the prevention of T2DM. Moreover, our umbrella review assessed the overlapping and excess significant finding among included meta-analyses. This provides evidence on the quality of previous reviews.

As a limitation of the study, we did not consider the primary individual included studies—instead, the results were considered only from SRMAs. Comparisons of effects among interventions is impossible. Moreover, the quality of the included SRMAs was low based on AMSTAR’s criteria [13]. This can lead to potential bias in the results of SRMAs. The overlapping in the included meta-analyses were mostly moderate and high, indicating that the included SRMAs could not much add in new evidence. In addition, only two databases without grey or unpublished literature were searched to identify the relevant studies. Therefore, publication bias might be presented in our review.

Most included studies in SRMAs were mainly from developed countries, the transferability of results into specific local context is questionable as the food components of certain diets such as Mediterranean diet are not often consumed in other regions. Therefore, future studies should consider the application and effects of these dietary patterns on other geographical areas. In addition, the results from excess significant findings showed that meta-analyses among this group might overestimate the real effect size of the intervention. Therefore, the interpretation of results in these meta-analyses should be cautioned. Given the fact that almost all reviews included in this umbrella review have low confidence in the evidence, an updated meta-analysis that adheres to the criteria of AMSTAR 2 is needed.

Other than diet interventions, pharmacological interventions have shown significant benefits in preventing T2DM especially in high-risk individuals such as people with IFG and IGT [53]. However, the scope of this review did not cover the efficacy of medication treatments and therefore, the comparative effectiveness between lifestyle and pharmacological interventions cannot be assessed in our review. Although, the results from previous systematic review and network meta-analysis that compared the efficacy among all available lifestyle and pharmacological interventions found that glipizide and diet intervention plus pioglitazone had the highest probabilities of being the most effective interventions for T2DM prevention [16]. Potential risk and benefit together with cost-effectiveness of drug treatments relative to lifestyle interventions should be considered when making clinical and policy recommendations.

## 5. Conclusions

In summary, healthy dietary patterns such as Mediterranean, DASH, and diet with the low glycemic index, diet interventions including low calorie and low-fat diets, and combined diet and physical activity interventions, and low consumption of red/processed meat, and sugar sweetened beverages significantly decreased the risk of T2DM. These lifestyle modifications should be promoted in both individual and population levels to prevent and decrease the burden of T2DM in the future.

## Figures and Tables

**Figure 1 nutrients-12-02722-f001:**
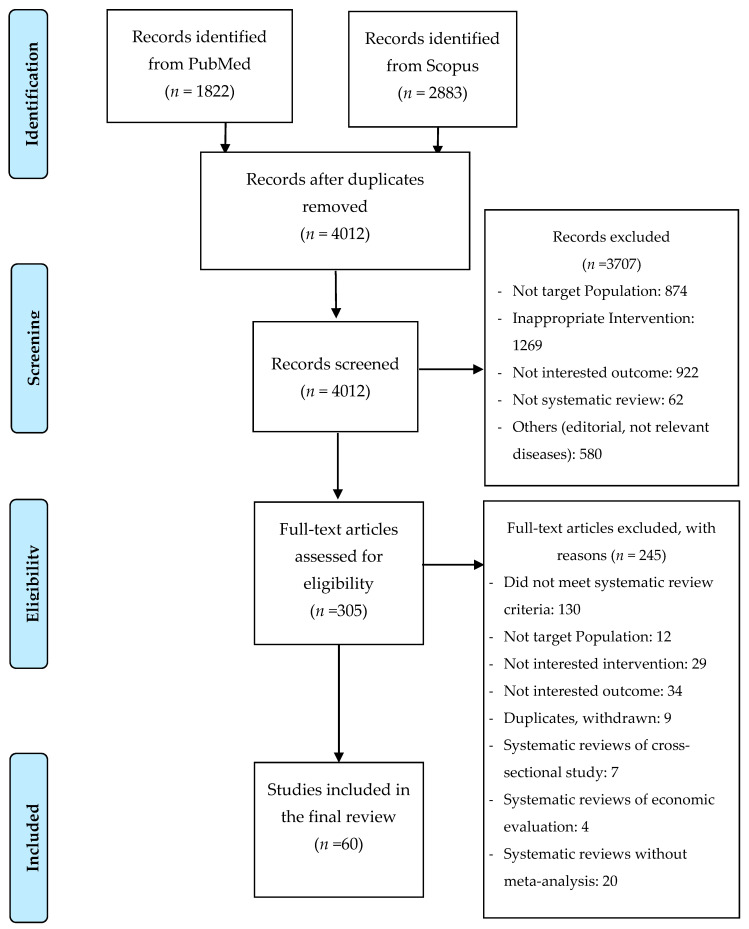
Flow diagram of study selection.

**Figure 2 nutrients-12-02722-f002:**
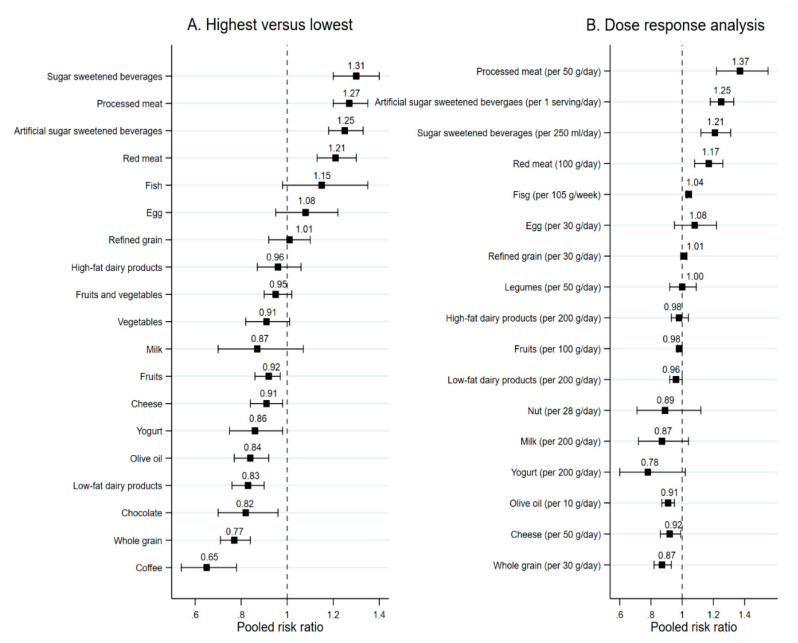
Pooled risk ratios of food groups and risk of T2DM. (**A**) Highest versus lowest category comparison. (**B**) Dose-response analysis.

**Figure 3 nutrients-12-02722-f003:**
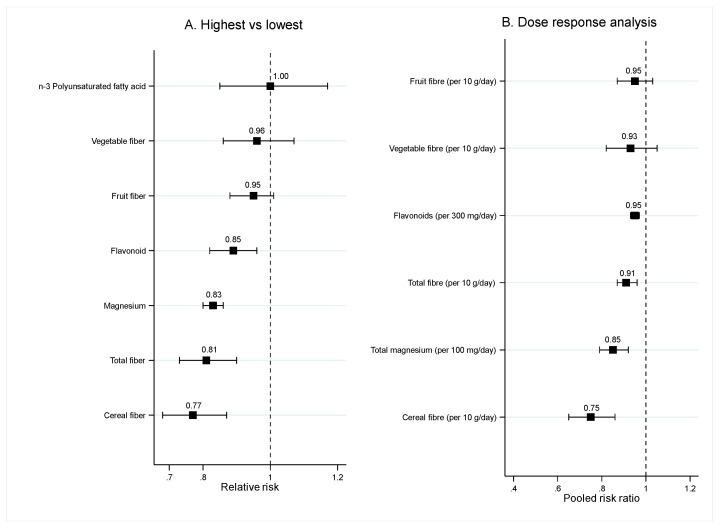
Pooled risk ratios of food nutrients and risk of T2DM. (**A**) Highest versus lowest category comparison. (**B**) Dose response analysis.

**Figure 4 nutrients-12-02722-f004:**
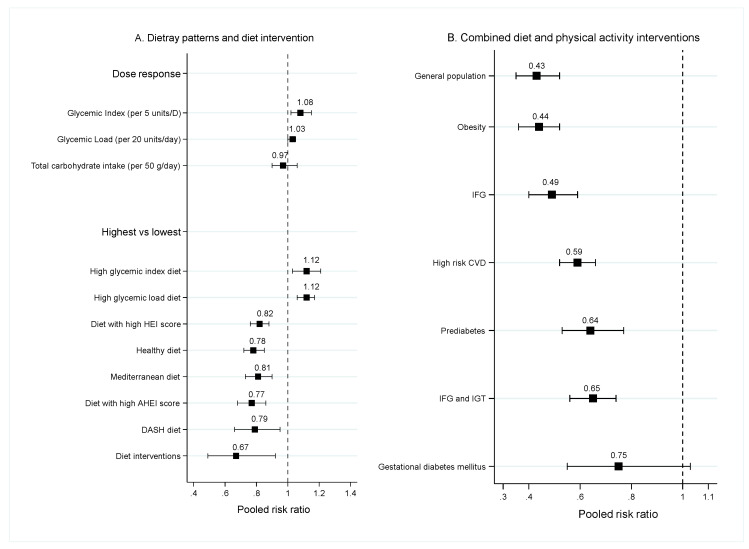
Pooled risk ratios of dietary patterns, diet and combined diet and physical activity interventions and risk of T2DM. (**A**) Dietary patterns and diet intervention. (**B**) Combined diet and physical activity interventions.

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
