# Peer review of "Preventive Role of Diet Interventions and Dietary Factors in Type 2 Diabetes Mellitus: An Umbrella Review"

_nutrients, 2020, doi:10.3390/nu12092722_

Round 1

Reviewer 1 Report

In this umbrella review of systematic reviews, the authors attempted to synthesize evidence on diet interventions and dietary factors in prevention of type-2 diabetes mellitus (T2DM),  and summarized the findings of multiple reviews appropriately. This is a through and well conducted review, which includes a quality assessment of the studies included. However, 2 main limitations are identified, databases were searched for relevant studies until June 2018. There is a need to update this search for the last two years.

The authors state that the results from this this review should be beneficial for policy makers in applying them in clinical and public health practices. However, for this interpretation there is lack of discussion about potential confusion factors in the relation between dietary factors and T2DM prevention including pharmacological treatment. The authors should refer to this in their results and discussion section.

Author Response

Reviewer 1

Point 1. In this umbrella review of systematic reviews, the authors attempted to synthesize evidence on diet interventions and dietary factors in prevention of type-2 diabetes mellitus (T2DM),  and summarized the findings of multiple reviews appropriately. This is a through and well conducted review, which includes a quality assessment of the studies included. However, 2 main limitations are identified, databases were searched for relevant studies until June 2018. There is a need to update this search for the last two years.

Response: We have updated our searching until 19th August 2020. We have found additional eight systematic reviews and meta-analyses that met our inclusion criteria. The manuscript has been revised by including the results of these additional systematic reviews, see Methods (page 2; line 82), Results (pages 3-10), and Figure 1, Figure 3 and Figure 4.

Point 2. The authors state that the results from this this review should be beneficial for policy makers in applying them in clinical and public health practices. However, for this interpretation there is lack of discussion about potential confusion factors in the relation between dietary factors and T2DM prevention including pharmacological treatment. The authors should refer to this in their results and discussion section.

Response: We have added and discussed the efficacy of pharmacological treatment compared with lifestyle interventions in the discussion part (page 12; lines 435-444).

Reviewer 2 Report

This study entitled "Preventive role of diet interventions and dietary factors in type 2 diabetes mellitus: An umbrella review" aimed to synthesize evidence on diet interventions and dietary factors in prevention of T2DM.

In general, it is a quite interesting study which bring new evidences. However, before publication, some aspects should be addressed.

ABSTRACT

- Conclusion should be more specific.

KEYWORDS

- Do not put as keyword the same words from the title.

INTRODUCTION

- References should checked. Line 54 are some errors.

METHODS

- My major concern is regarding the data of the search. June 2018 is more than 2 year ago. The search must be updated. As you have the search strategy, is just run from June/2018 to July/2020.

- The fact of just two data base were addressed must be included as a limitation from this study.

- Selection of studies (line 83-89): it is important state clear if the study have some age group restrict or all ages could be included.

- Did you run any concordance test for the screening process between both evaluators? For example concordance index (%) or kappa test?

RESULTS AND DISCUSSION

- Well written section. Congrats.

Author Response

Reviewer 2

This study entitled "Preventive role of diet interventions and dietary factors in type 2 diabetes mellitus: An umbrella review" aimed to synthesize evidence on diet interventions and dietary factors in prevention of T2DM.

In general, it is a quite interesting study which bring new evidences. However, before publication, some aspects should be addressed. 

Point 1. ABSTRACT

- Conclusion should be more specific.

 Response: We have revised the abstract to make it more specific (see page 1).

Point 2. KEYWORDS

- Do not put as keyword the same words from the title.

Response: We have revised the keywords according to the comment (see page 1).

Point 3. INTRODUCTION

- References should checked. Line 54 are some errors.

Response: We have corrected this mistake (page 2; line 55).

METHODS

Point 4. My major concern is regarding the data of the search. June 2018 is more than 2 year ago. The search must be updated. As you have the search strategy, is just run from June/2018 to July/2020.

Response: We have updated the searching until 19th August 2020. Eight additional studies have been eligible and included in the revised manuscript, see Methods (page 2; line 82) and Figure 1.

Point 5. The fact of just two data base were addressed must be included as a limitation from this study.

Response: Limitation of using only two databases has been stated in the discussion part (see page 12; lines 424-425).

Point 6. Selection of studies (line 83-89): it is important state clear if the study have some age group restrict or all ages could be included.

Response: We have specified the specific age group of study’s participants in the inclusion criteria according to the comment (see page 2; line 89).

Point 7. Did you run any concordance test for the screening process between both evaluators? For example concordance index (%) or kappa test?

Response: We estimated the percent agreement between two reviewers (92.68%) and kappa statistic (0.83). Results of these have been presented in the Method part (see page 2; lines 86-87).

Point 8. RESULTS AND DISCUSSION

- Well written section. Congrats.

Response: Thank you very much for your comments.

Reviewer 3 Report

Minor comments are as following for the authors to consider for revising:

  • Please check the English spellings including, 'level pf physical activity', the 2nd para, page 2; 'risk ration', the first para, page 3; and 'fisg', figure 2, page 5.
  • Could please provide the full name of short form 'SR' in the figure 1, page 4, and 'CCA', the 2nd para, page 8?
  • In the 2nd column at right side in figure 1, the sum of 'full-text articles excluded without reasons' seems 211, not 207. In addition, both in right columns, 'not target population' is indicated. It would be better to  specify study populations in the supplementary tables?
  • Were there any considerations of the publication bias of this review? If so, the authors might mention them in Discussion.

Author Response

Reviewer 3

Minor comments are as following for the authors to consider for revising:

Point 1. Please check the English spellings including, 'level pf physical activity', the 2nd para, page 2; 'risk ration', the first para, page 3; and 'fisg', figure 2, page 5.

Response: We have corrected these wrong spellings.

Point 2. Could please provide the full name of short form 'SR' in the figure 1, page 4, and 'CCA', the 2nd para, page 8?

Response: We have stated the full names as suggestion.

Point 3. In the 2nd column at right side in figure 1, the sum of 'full-text articles excluded without reasons' seems 211, not 207. In addition, both in right columns, 'not target population' is indicated. It would be better to  specify study populations in the supplementary tables?

Response: We are sorry for this mistake. The numbers of studies in Figure 1 have been revised according to the numbers of studies identified from updated searching. Types of study populations were stated in Supplementary Table 3.

Point 4. Were there any considerations of the publication bias of this review? If so, the authors might mention them in Discussion.

Response: We did not search the grey or unpublished literatures. Therefore, publication bias might be presented in our study. This limitation has been mentioned in the discussion part (see page 12; lines 424-425).

Round 2

Reviewer 2 Report

Congrats for the authors.

All my concerns were correctly addressed.